# Pneumoviral Phosphoprotein, a Multidomain Adaptor-Like Protein of Apparent Low Structural Complexity and High Conformational Versatility

**DOI:** 10.3390/ijms22041537

**Published:** 2021-02-03

**Authors:** Christophe Cardone, Claire-Marie Caseau, Nelson Pereira, Christina Sizun

**Affiliations:** CNRS, Université Paris-Saclay, Institut de Chimie des Substances Naturelles, 91198 Gif-sur-Yvette, France; cardone.christophe@gmail.com (C.C.); cmcaseau@gmail.com (C.-M.C.); nelson.matthieu.pereira@gmail.com (N.P.)

**Keywords:** respiratory syncytial virus, metapneumovirus, *Pneumoviridae*, RNA polymerase, phosphoprotein, intrinsically disordered protein, protein folding, tertiary complex, multidomain protein

## Abstract

*Mononegavirales* phosphoproteins (P) are essential co-factors of the viral polymerase by serving as a linchpin between the catalytic subunit and the ribonucleoprotein template. They have highly diverged, but their overall architecture is conserved. They are multidomain proteins, which all possess an oligomerization domain that separates *N*- and *C*-terminal domains. Large intrinsically disordered regions constitute their hallmark. Here, we exemplify their structural features and interaction potential, based on the *Pneumoviridae* P proteins. These P proteins are rather small, and their oligomerization domain is the only part with a defined 3D structure, owing to a quaternary arrangement. All other parts are either flexible or form short-lived secondary structure elements that transiently associate with the rest of the protein. *Pneumoviridae* P proteins interact with several viral and cellular proteins that are essential for viral transcription and replication. The combination of intrinsic disorder and tetrameric organization enables them to structurally adapt to different partners and to act as adaptor-like platforms to bring the latter close in space. Transient structures are stabilized in complex with protein partners. This class of proteins gives an insight into the structural versatility of non-globular intrinsically disordered protein domains.

## 1. Introduction

Among multidomain proteins, those with very large intrinsically disordered regions form a special class, as these regions can delineate domains of their own. Disordered regions are generally associated with a low complexity, as they form neither stable secondary nor tertiary structures [1,2]. As linkers, they allow the relative positioning of globular domains. Their plasticity comes into play for repositioning adjacent globular domains under different conditions. They are also often involved in signaling and regulation by post-translational modifications. However, large intrinsically disordered regions may be more than adaptable linkers. Here, we present the structural properties of a family of nearly fully intrinsically disordered proteins, namely the phosphoproteins of pneumoviruses. Their domain organization is fundamental for their function as essential polymerase co-factors for these viruses. We show how structural complexity is gained by the presence of a very short oligomerization domain, which results in quaternary organization, and how the plasticity of the proteins leads to very different structures in a variety of protein complexes. Despite or rather because of their low intrinsic structural complexity, these P proteins are puzzling multifunctional proteins that can engage in multiple interactions with weak to high affinities. We review here the structural specificities of the *Pneumoviridae* P proteins, using RSV and hMPV P as examples. We focus on the multiplicity of binding sites afforded by the different domains, either on a single domain or consecutive domains, and show how they locally switch conformations when engaged in protein complexes. This ability confers them adaptor-like properties, since they position protein partners, which do not directly bind to each other, close in space to form functional complexes.

## 2. Structural Features of *Mononegavirales* Phosphoproteins

Human respiratory syncytial virus (hRSV) and human metapneumovirus (hMPV) are enveloped viruses of the *Mononegavirales* order [3]. They are the archetypes of *Orthopneumovirus* and *Metapneumovirus* genera inside the *Pneumoviridae* family [4]. Pneumoviruses, i.e., viruses of this family, are major pathogens for pediatric acute respiratory disease, worldwide. hRSV is the main agent of acute lower respiratory infections in infants and young children, causing pneumonia and bronchiolitis, and thus a leading cause for pediatric hospitalization and child mortality [5,6]. The bovine form of RSV is a major pathogen for bovine respiratory disease in young calves and represents an important economic burden for the cattle industry [7]. hMPV is the second most common cause, after hRSV, of acute respiratory disease and related hospitalization of young children [8,9].

The viral genome of *Mononegavirales* is a ribonucleoprotein complex (RNP) consisting of a non-segmented, negative-strand RNA molecule that is encapsidated by the viral nucleoprotein (N) [10,11,12,13]. The RNP complex serves as the template for the viral RNA polymerase complex during transcription and replication, which produce mRNA and genomic RNA, respectively. While mRNA remains naked, newly synthesized genomic RNA is covered by N as it is synthesized. The apo polymerase complex is formed a minima by the large polymerase unit (L), a protein endowed with enzymatic activities, and its essential co-factor, the viral phosphoprotein (P) [14,15]. In the holo RNA polymerase complex, the P protein acts as a linchpin that tethers and positions the polymerase on the RNP, by binding to both L and N. It has been suggested that P is involved in local decapsidation of the viral genome to allow L access to the genomic RNA [16]. Due to their multimeric organization, it has been suggested that *Mononegavirales* P proteins bind to consecutive N units on the RNP and that P proceeds by cartwheeling along the RNP for RNA elongation by L [10].

While the *Mononegavirales* L proteins are structurally conserved [15], the *Mononegavirales* P proteins display pronounced structural diversity. Their lengths vary considerably: RSV P is the shortest with 241 residues, and Nipah virus P the longest with 709 residues. P proteins do not share sequence similarity beyond the level of subfamilies or families within the *Mononegavirales*. However, they display the same overall architecture and common structural features. First, they contain a central oligomerization domain, delimiting the *N*- and *C*-terminal domains. The oligomerization domain contains a helical coiled-coil core of variable arrangement and length (Figure 1). The P oligomerization domains are arranged either as parallel dimers, such as in the vesicular stomatitis virus [17] or rabies virus P [18]; as antiparallel dimers of parallel dimers, such as in the mumps virus P [16]; as parallel tetramers, such as in the measles virus [19] or in hMPV P [20]; or parallel trimers, such as in some *Filoviridae* virus phosphoproteins [21,22] (Figure 1).

A second common feature of the *Mononegavirales* P proteins is the presence of large intrinsically disordered regions, on each side of the oligomerization domain [23,24,25]. Some of these P proteins possess small folded domains, involved in binding to N complexed to RNA (N-RNA complex), located at the *C*-terminus, such as the *C*-terminal X domain of the *Paramyxoviridae* P proteins formed of a 3-helix bundle [26] or the *C*-terminal domain of the rabies virus [27,28] (Figure 1). In the case of *Filoviridae* phosphoproteins, a folded *C*-terminal domain is involved in specific interferon inhibition [29]. The structural differences point to the evolutionary divergence of the *Mononegavirales* P proteins, which have evolved specific interactions with their cognate N and L proteins, and thus specific RNP recognition mechanisms by the cognate polymerase. In the case of *Pneumoviridae* P proteins, the oligomerization domain delimits two intrinsically disordered domains, denoted here P_NT_ and P_CT_ (Figure 1).

## 3. The Three Domains in *Pneumoviridae* Phosphoproteins Display Multiple Binding Sites Associated to Multiple Functions

RSV and hMPV P are composed of 241 and 294 amino acids, respectively, and have been both reported to form tetramers [20,30,31]. Sequence alignment of the *Pneumoviridae* phosphoproteins, adjusted for structural similarity on the basis of the recently reported L–P complex structures [32,33,34], points to two regions with marked conservation: the oligomerization domain and a proximal *C*-terminal region inside the P_CT_ domain (Figure 2). In the P_NT_ domain, the *N*-terminus and the region proximal to the oligomerization domain also display conservation, but specificity remains for the *Orthopneumovirus* and *Metapneumovirus* genera. The three domains, P_NT_, P_OD_ and P_CT_, are involved in specific complexes associated to different functions of P, involving mainly P regions with a certain degree of conservation.

As the main co-factor of the viral RNA polymerase, P binds to L via nearly the entire P_CT_, with the exception of a *C*-terminal tail, completed by P_OD_ [32,33,34,36,37]. Recognition of the RNP template by the RNA polymerase is mediated by a direct interaction between the P and N proteins, also via P_CT_ [38]. In this N–P binding mode, the *C*-terminal tail of P is the primary binding region for the RNP and more generally a binding site for the RNA-bound N [38,39,40] (Figure 2). The L–P complex must be of high affinity, whereas binding of P to the RNP must be sufficiently weak to allow processivity of the polymerase. The N–P interaction is also required for the formation of cytoplasmic virus-induced, highly dynamic organelles, termed inclusion bodies (IBs), which act as viral factories, where N, L and P proteins accumulate [38,41,42,43,44,45].

Like all *Mononegavirales* P proteins, *Pneumoviridae* P proteins may also engage with their cognate N protein following a second binding mode, which is distinct from the N-RNA–P binding mode. In this mode, P serves as a chaperone for N, by maintaining neo-synthesized N monomeric and mRNA-free, and thus competent for encapsidation of genomic RNA. The interaction with N in the N^0^–P mode is mediated by a conserved region located in P_NT_ [11] (Figure 2).

The viral transcription activity of the *Pneumoviridae* is modulated by an additional co-factor, the M2-1 phosphoprotein. RSV M2-1 is required for efficient transcription of RSV by preventing premature transcription termination at the gene junctions [46,47,48]. For hMPV, it was reported to be dispensable [49]. In infected cells, M2-1 is recruited to the viral cytoplasmic IBs [43,50], via a direct interaction with RSV P_NT_ [51,52] (Figure 2). Dynamic phosphorylation/dephosphorylation of RSV M2-1, and speculatively MPV M2-1, is needed for efficient transcription. Dephosphorylation of RSV M2-1 is operated by the cellular PP1 phosphatase, which binds to P in a conserved region proximal to the M2-1 binding region [52] (Figure 2). It thus appears that the specific role of *Pneumoviridae* P for transcription emanates from P_NT_.

RSV P was found to be constitutively phosphorylated in vivo [53,54]. Two main clusters of phosphorylated serines, S116/S117/S119 and S232/S237, were identified in the P_NT_ and P_CT_ domains (Figure 2) [55,56,57]. The phosphorylation state of these serines was reported to modulate RSV replication and budding in vivo [55]. P phosphorylation was also shown to regulate several protein–protein interactions. RSV P mutants impaired for phosphorylation at the main sites displayed reduced interaction between the N and P proteins and resulted in attenuated RSV virus or impaired budding [55]. However, most constitutive phosphorylation sites were reported to be dispensable for viral RNA transcription and virus replication in vitro [55,57,58]. Moreover, among the phosphorylatable serine/threonine residues in RSV P, only position 119 seems to be conserved in the *Pneumoviridae* family. The S/T119-F/Y-D/E-E consensus sequence corresponds to the general S-X-X-E/D recognition motif of casein kinase II (Figure 2). Otherwise, sequence alignment shows that the phosphorylation sites are located in variable regions (Figure 2), suggesting that they are specific for each virus or genus. Notably phosphorylation of RSV T108 prevented interaction with RSV M2-1 [52,59], but T108 has no counterpart in MPV.

## 4. *Pneumoviridae* Phosphoproteins Display a High Degree of Intrinsic Disorder

### 4.1. The Oligomerisation Domain Is the Only Stably Folded Domain in RSV/hMPV Phosphoproteins

Whereas high-resolution structural information and structural models could be obtained for the RSV and hMPV N proteins in the form of N^0^, N-RNA rings or nucleocapsids [40,60,61] the P proteins have long eluded detailed structural elucidation. A combination of computational, biochemical and biophysical experiments, performed by different laboratories, pointed to a similar domain organization, with a central oligomerization domain that is critical for the function of P [30,31,58,62,63,64]. Mass spectrometry in combination with trypsin degradation assessed the overall intrinsic disorder in RSV P [62,65].

The oligomerization domain of RSV P was first mapped to residues Y120–H150 by cross-linking and deletion mutants [30]. The approximate location of the oligomerization domain was confirmed by trypsin-chymotrypsin digestion, resulting in the Y* fragment S119–T160 [31], and by the reduced oligomerization capacity of the double mutants L135A/L142A and D139K/E140K [58]. Cross-linking [30] and sedimentation experiments [31] indicated a stoichiometry of four protomers in the oligomerization domain. Electron micrographs of negatively stained P preparations revealed objects with an elongated shape [31]. The high stability of the oligomeric state of P was assessed by denaturation experiments, while the helical content of the oligomerization domain was established by far UV circular dichroism [63].

Sequence alignment indicates that the RSV P region R134–I178 (R175–I219 in hMPV P), comprising the oligomerization domain P_OD_, has the highest conservation among the *Pneumoviridae* P proteins (Figure 2). High-resolution structural information obtained from the first X-ray crystallography structure of the hMPV oligomerization domain [20] was therefore expected to be transposable to RSV P with high confidence. This hMPV P structure confirmed the tetrameric stoichiometry and revealed the quaternary arrangement of a rather small region (residues S171–N194, equivalent to N130–V153 in RSV P) into a parallel helical coiled-coil. Two other crystal structures of hMPV P_OD_ displayed a similar arrangement, but revealed relative flexibility as well as fraying at the *C*-terminal end of the coiled-coil helices [66], suggesting that it may be subject to structural variations. Interestingly, the crystals yielding the hMPV P_OD_ structure were obtained from the hMPV P construct S158–E237, containing large *N*- and *C*-terminal regions flanking P_OD_, which probably underwent proteolysis, confirming the intrinsic disorder outside of P_OD_.

### 4.2. Relative Disorder in the N- and C-terminal Regions of RSV/hMPV Phosphoproteins

Although intrinsically disordered proteins and large intrinsically disordered regions do not adopt a defined secondary or tertiary structure [67], they nonetheless display complexity in the form of structural and temporal heterogeneity [1,2]. *Pneumoviridae* P proteins provide good illustrations for these properties. Different biophysical approaches, performed in solution and combined with a fragment approach, where several P constructs devoid of *N*- or *C*-terminal regions or P_OD_ were used, gave insight into the differential dynamics and secondary structure propensities of the RSV and hMPV P proteins.

The overall secondary structure propensity and compactness of RSV P was assessed by circular dichroism and size exclusion chromatography, and pointed to a modular behavior of P [63,64]. The oligomerization domain was singled out as a stably folded α-helical domain. However, thermal and chemical denaturation assays conducted on RSV P and several complementary P deletion constructs also indicated α-helical propensity outside of the oligomerization domain, in more loosely packed regions [63,64]. Whereas the P_NT_ region was found to be fully disordered, P_CT_ displayed α-helical propensity in the S161–R197 region, located *C*-terminally to P_OD_ (Figure 2). The secondary structure propensity of this *C*-terminal module could be modulated, e.g., by changing the pH, with a gain in α-helicity correlated with increased compaction. These results were interpreted in terms of folding into a pre-molten globule, displaying a reversible order to disorder transition in the 5–37 °C range, with low but still significant cooperativity [64].

The conformational distribution of hMPV P in solution was investigated by small angle X-ray scattering (SAXS) and molecular dynamics, also by adopting a fragment approach [20,66]. Structural ensembles, selected by adjusting the SAXS curves, contained α-helices *C*-terminally to the oligomerization domain. In contrast to the coiled-coil oligomerization domain, they displayed inter-protomer mobility and structural lability [20,66]. In some structural models, they packed against the oligomerization domain, resulting in the compaction of P. However, these interactions appeared to be rather weak, according to chemical denaturation experiments followed by SAXS. Helical propensity was also predicted at the *N*-terminus of hMPV P, in the G13–R28 stretch (equivalent to G10–K25 in RSV P) of a fragment encompassing the first 60 residues [66]. This stretch coincides with the hMPV N^0^ binding site [60], which displays high sequence conservation among *Pneumoviridae* P proteins (Figure 2). Finally, α-helical propensity was predicted for a second region in hMPV P, E253–V262 (equivalent to P218–L227 in RSVP) [66], which is part of the hRSV L binding site [36].

### 4.3. Characterization of Disorder at the Residue-Level of RSV Phosphoprotein

Whereas SAXS proved to be a powerful tool to investigate the overall conformational space and spatial extension of hMPV P, as well as its intrinsically disordered regions in solution, Nuclear Magnetic Resonance (NMR) gave detailed structural and dynamic insight at an atomic scale for the RSV P protein. Different experiments were performed on the RSV P to probe its secondary structure propensity, local flexibility and spatial proximity between the protein regions, at the single residue level [68]. Most of these experiments rely on the acquisition of 2D ^1^H-^15^N heteronuclear single quantum correlation (HSQC) spectra, which display correlation peaks for the amide ^1^H-^15^N pairs, as shown in Figure 3. Sharp signals with low chemical shift dispersion in the ^1^H dimension were observed, indicating a large proportion of intrinsically disordered regions for RSV P. However, neither the oligomerization domain nor the proximal *C*-terminal α-helical module identified in [64] could be observed in the ^1^H–^15^N HSQC spectra, due to unfavorable dynamics.

Finer analysis was done by investigating the per residue secondary structure propensity (SSP) of RSV P by NMR, based on backbone chemical shifts, also by adopting a fragment approach [68]. Removal of the oligomerization domain in the P_ΔOD_ construct restored the signal in the *C*-terminal α-helical module, without significantly affecting the other signals ([68], Figure 3). SSP predictions, made with different algorithms, provided qualitatively similar SSPs, but differed in quantification (Figure 4). A transient helix, with a maximal propensity of 10–20 %, was detected for RSV P_NT_ residues ~D12–I24 [68] (Figure 4). α-helical propensity of the same order (~20%) was also reported in the P_NT_ of hMPV P, residues M1–K60, by fitting SAXS data with an ensemble of structural models [66]. In RSV P_ΔOD_ and other P fragments devoid of the oligomerization domain, the region T160–T210 displayed a rather high α-helical propensity, with two adjacent transient helices (Figure 4). The SSP profiles also suggested that several regions might contain short strands as well as additional extremely transient α-helices. Dihedral angle ranges were derived from backbone chemical shifts for the P regions with a high secondary structure propensity and were used as constraints to build the structural models shown in Figure 4.

^15^N spin relaxation [72,73,74] was used to get deeper insight into the protein backbone dynamics. The relaxation parameters, such as longitudinal (R_1_) and transverse (R_2_) ^15^N spin relaxation rates, as well as the amide ^1^H–^15^N steady state nuclear Overhauser effects (NOE) were measured for different deletion constructs of P [68] and for full-length P (Figure 5). These parameters were not homogeneous along the amino acid sequence of P. The ^1^H–^15^N NOE values were all below 0.5, suggesting that internal motions took place at the ps-ns timescale. This is in line with the overall disordered state of RSV P outside of the oligomerization domain. Regions with α-helical propensity displayed the highest ^1^H–^15^N NOE values, suggesting that these regions were less flexible. The concomitant increase of R_2_ and decrease of R_1_ in the α-helical *C*-terminal module in P_ΔOD_ indicated slower tumbling, and confirmed the high secondary structure propensity of this region. Increased R_2_ values in several regions of the P_NT_ domain pointed to rapid conformational exchange, on the µs-ms timescale, between a disordered state and a minor α-helical state. Interestingly, the ^15^N spin relaxation revealed regions that underwent conformational exchange within the P_NT_ domain, for which the α-helical SSP could not unambiguously be determined, e.g., the binding site for RSV M2-1 [52] (Figure 2, Figure 4 and Figure 5).

Finally, incorporation of nitroxide spin labels at given positions [75] was used to probe transient long-range contacts within a 15 Å distance in full-length RSV P by paramagnetic relaxation enhancement experiments [68]. This type of experiment showed that the oligomerization domain as well as all short-lived secondary structure elements transiently associated with each other, at the hydrophobic surface patches exhibited in the stabilized α-helices. This network of transient interactions would explain the overall compaction observed for P in the absence of stably folded domains [64].

### 4.4. Disorder Correlates with Fuzzy Binding of the Phosphoprotein to the Nucleoprotein

The interaction between P and N is fundamental for viral replication in *Mononegavirales* as it drives recognition of the RNP by the RNA polymerase. The primary binding region for N in RSV P was mapped to a 9-residue-long *C*-terminal tail, which is in a fully disordered state when P is unbound [38,39] (Figure 4 and Figure 5). The core of the *Mononegavirales* and *Pneumoviridae* N proteins is formed of two *N*- and *C*-terminal domains, N_NTD_ and N_CTD_, held together by a hinge, where the genomic RNA is tightly bound [11,24]. The N_NTD_ domain alone does not bind RNA and can be produced as a monomeric, RNA-free recombinant protein. However, it retains the ability to bind the *C*-terminus of P and can be used as a model to investigate the N-RNA- or RNP-binding mode [76]. A 30 µM in vitro binding constant was measured for a complex between P_CT_ and N_NTD_ by surface plasmon resonance [76]. This value is compatible with the requirement for polymerase processivity.

Structural determinants for N–P binding were obtained by X-ray crystal structures of the N_NTD_ co-crystallized with phenylalanine and a series of peptides containing the *C*-terminal of P, of 2 up to 13 residues [76]. The *C*-terminal aromatic P F241 residue was consistently found inserted in a hydrophobic pocket at the surface of N_NTD_, providing the anchor point of P. For short P peptides, the D240 residue could also be observed, but all other residues appeared to be highly disordered. This raised the question if only residues binding to a defined and ordered site were sufficient to account for the whole binding process.

An answer was provided by NMR. Chemical shift perturbations induced by P peptides were observed in ^1^H–^15^N HSQC spectra of the ^15^N-labeled N_NTD_ for residues that were inside but also outside the P binding pocket [76]. When using the ^15^N-labeled P_CTD_ fragment, spectral perturbations were observed for the last 10 *C*-terminal residues in the presence of N_NTD_ [68]. Altogether these results suggested that P binding to N was not restricted to the D240 and F241 residues. This was corroborated by affinity measurements conducted with NMR titration experiments. They indicated that the binding strength of the P peptides increased with their length: the dissociation constants were ~5 mM for the D240–F241 dipeptide, and 25–50 µM for a 12-amino-acid-long P peptide and for P_CTD_, respectively [68,76]. These findings suggested that affinity is driven by additional interactions, mediated by the full *C*-terminal tail of P. This contains many negatively charged residues that bind in a fuzzy fashion to the positively charged surface of N outside of the F241-binding pocket [77,78].

A proof of concept for the druggability of this interaction was given with a small molecule that competed with P for binding and displayed some antiviral activity with a recombinant RSV virus [76]. The scaffold of this molecule was based on an aromatic moiety and a carboxylate mimicking the F241 and D240 side chains. Interestingly, none of the two *C*-terminal residues are conserved between the RSV and MPV viruses (Figure 2). The *C*-terminal tails are highly acidic for both proteins, but have very different lengths, with 9 and 26 amino acids for RSV and MPV P, respectively, indicating that even fuzzy binding modes are specific for each virus type.

## 5. Folding Upon Binding in the Intrinsically Disordered Domains of *Pneumoviridae* Phosphoprotein 

### 5.1. The Disordered N-Terminus of P Folds in the N^0^–P Complex

In addition to the N–P binding mode described above, a second N^0^–P binding mode corresponds to the chaperone activity of P. The N^0^–P complex was investigated for both RSV and hMPV [60,79]. This complex is tighter than the labile N_NTD_–P_CT_ complex. A 5 µM affinity was measured by surface plasmon resonance between RSV P_NT_ and a monomeric N K170A/R185A mutant, impaired for RNA binding [79]. For RSV, the N^0^-binding region was mapped to residues M1–T29 by mutational analysis and validated with a minigenome [79].

A simplified RSV N^0^–P complex was reconstituted using an *N*-terminally truncated form of N, impaired for oligomerization, and an *N*-terminal P peptide of 40 residues, but could not be crystallized [80]. A similar complex was obtained for the hMPV N^0^–P complex, by using a trypsin-treated fusion protein comprising a 40-residue *N*-terminal P peptide and full-length hMPV N. The structure of the latter was solved by X-ray crystallography, giving high resolution insight into this binding mode [60]. The first 28 residues were well resolved, and residues G13–L27 (equivalent to G10-I24I in RSV P) formed an α-helix (Figure 6). The latter region displays a high degree of sequence conservation (Figure 2), which suggests that the transient helix observed in free RSV P can be stabilized upon complex formation (Figure 4). This was further corroborated by binding assays of the RSV *N*-terminal P peptides, spanning residues E11–S30 and constrained into an α-helical conformation by stapling to the monomeric N. These stapled peptides competed with the equivalent wild-type peptide and furthermore exhibited antiviral activity on recombinant RSV viruses in vitro and in vivo [81].

### 5.2. Pneumoviridae P Protomers Each Adopt a Unique Fold in the L–P Complex

Before high-resolution structural information became available for the L–P complex, two regions of P were reported to be the RSV L binding sites. A first region was mapped to residues Y121–T160, i.e., approximately to P_OD_, by using deletion mutants in co-immunoprecipitation assays [37]. When RSV L was produced as a recombinant protein, a second region in P_CT_, encompassing residues D216–N234, was determined [36]. According to NMR data on RSV P, this region did not display any α-helical propensity.

Very recently a major breakthrough was achieved with the cryogenic electron microscopy (cryo-EM) structures of hRSV and hMPV RNA polymerases in the elongation mode, featuring the complex between the L and P proteins [32,33,34]. These structures showed that P engaged to L via P_OD_ and nearly the full P_CT_ (Figure 6), confirming the previously determined regions [37]. Although the complexes were produced with full-length P, the density for P_NT_ was missing, showing that P_NT_ was not directly involved in L-binding [32,33,34]. Moreover, these structures confirmed that the oligomerization domain of P was organized in a parallel coiled-coil helix bundle, with a core spanning residues N131–T151 in RSV P. Interestingly, three protomers displayed an extension of the oligomerization domain α-helices by six residues, whereas one is prolonged by a β-strand. In free P, this heterogeneity is manifested by fraying at the *C*-terminus of these helices. The most striking aspect of these complexes is that each P protomer in P_CT_ adopts a distinct conformation, consisting of helices of different boundaries and lengths. It is the same in the two RSV L–P complexes and very similar for hMPV L–P (detailed in Figure 2). This highlights the plasticity of *Pneumoviridae* P_CT_, required in this type of complex. The high helical propensity observed by NMR for free RSV P in the R173–K205 region shows that the α-helical structures in P_CT_ are already sampled in the free form. They are stabilized, each in a specific way, by binding to L. Only the small most *C*-terminal α-helix L216–S232 in RSV P (equivalent to L251–T2267 in hMPV), observed for one protomer in the L–P complex, had not been detected in free P, suggesting that it folded upon binding to L. Interestingly, the only part of P_CT_ that does not engage with L are the 9 *C*-terminal residues forming the N-RNA binding site, suggesting that the N- and L-binding sites are proximal, but do not overlap. However, they are well positioned to bring the RNP template into close proximity to the polymerase, illustrating how P acts as an adaptor protein within the *Pneumoviridae* replication complex. The striking difference in size between the binding sites for the two partner proteins (up to 90 residues per protomer for L versus 9 residues for N) reflects the difference in binding strength, where P stays bound to L while it switches from one N protein to the next along the RNP template to allow elongation of the newly synthesized RNA molecule.

## 6. *Pneumoviridae* P Protein Serves as a Docking Platform for a Tertiary Complex Involved in Transcription Regulation

In line with the large intrinsic disorder in *Pneumoviridae* P proteins, several binding sites for partner proteins are localized to short linear motifs (SLiMs), which are short stretches in the protein sequences (Figure 2). Transient structures and weak binding are characteristic of these elements [67]. In eukaryotes, SLiMs are involved in cellular signaling [67,82,83]. Some viruses mimic eukaryotic SLiMs to interfere with the host immune response [82]. Viral SLiMs may also serve as recognition motifs between viral proteins. In RSV and hMPV P protein, the N^0^-binding region can be considered as an α-helix SLiM. Another SLiM was identified *N*-terminally to RSV P_OD_, as a binding site for transcription cofactor M2-1.

### 6.1. A Disordered Short Linear Motif in P_NT_ Folds Into an α-Helix in the RSV M2-1–P Complex

RSV M2-1 was early described as an RNA-binding protein [84,85]. It is a well-folded protein organized as a tetramer. Its 3D structure was solved for hRSV [86]. That of hMPV M2-1 was solved in complex with RNA [87]. Moreover, RSV M2-1 was shown to bind to P, in competition with RNA, on adjacent sites. The interaction between P and M2-1 is essential for RSV transcription [51,88,89], as P recruits M2-1 to the viral IBs [50,51]. An 8 nM affinity was measured by fluorescence anisotropy for full-length proteins [63]. When using a truncated form of M2-1, containing only the central domain of M2-1, a weaker binding of 3 µM was measured by isothermal titration calorimetry [51].

The binding domain for RSV M2-1 on P was mapped to the D100-Y120 region, just upstream of the oligomerization domain [88]. A P fragment encompassing residues P93–T110 was shown to be the smallest fragment able to bind to M2-1 [52]. The binding site was finely adjusted to a SLiM, residues F98–F109, which corresponds to an extremely transient α-helix in the free form [52] (Figure 4 and Figure 5). Three hydrophobic or aromatic P residues (L101, Y102 and F109), conserved in the *Orthopneumovirus* genus, were reported to be required for RSV M2-1 binding and in vitro transcription [88]. Information obtained by mutagenesis and NMR was used to model the RSV M2-1–P complex by docking the F98–F109 SLiM in α-helical conformation onto M2-1 [52]. This model was subsequently confirmed by structure resolution of this complex by X-ray crystallography [90] (Figure 7).

In the case of hMPV, there are no data for the M2-1–P complex. Although the M2-1 binding region displays some sequence conservation between the *Orthopneumovirus* and *Metapneumovirus* P proteins (Figure 2), only one critical residue (L101) is conserved. Moreover, hMPV M2-1 is not essential for the processivity of transcription, in contrast to RSV [49]. This suggests that the M2-1–P complex is specific to each genus in the *Pneumoviridae* family, possibly with distinct binding sites. The hMPV P protein should be amenable to NMR studies in the same way as hRSV P. This would provide residue-specific interaction information and allow a more precise structural comparison between these two proteins.

### 6.2. A Short Linear Motif in RSV P_NT_ Binds to the PP1 Phosphatase That Dephosphorylates M2-1

The bulk phosphorylation of M2-1 or P does not play a major role in the RSV M2-1–P interaction [52,88]. Although M2-1 is mainly unphosphorylated in RSV-infected cells [84], it needs nevertheless to be phosphorylated for efficient transcription [85]. For hMPV, M2-1 phosphorylation upregulates viral replication [92]. The phosphorylation state of RSV M2-1 determines its localization and mRNA-binding properties. Unphosphorylated M2-1 concentrates in the dynamic sub-compartments of viral IBs, where viral mRNA is produced and accumulates [43,52]. Dephosphorylation of M2-1 is achieved by the cellular phosphatase PP1, which is co-localized with M2-1 in RSV IBs [52]. Importantly, M2-1 and PP1 do not interact directly, but PP1 binds to P via an RVxF-like consensus motif present in the P protein [52].

RVxF PP1-binding motifs are found in intrinsically disordered regions of PP1-binding proteins [93]. In *Pneumoviridae* P proteins, such a motif is located in P_NT_, *N*-terminally to the M2-1-binding motif. The K82–F87 (K123–F127 in hMPV) stretch is very well conserved throughout the *Pneumoviridae* members (Figure 2). It is a good illustration for a viral mimetic of a eukaryotic SLiM. A homology model built for the PP1–P complex, using the X-ray crystallographic structure of a complex between the catalytic unit of PP1 and the RVxF motif of another PP1-binding protein, Gadd34, gives insight into the binding mode. The RVxF-like motif in RSV P is expected to adopt an extended structure, which is already sampled in the free form, as seen from the secondary structure propensities (Figure 4), and laterally docks onto a β-sheet in PP1 (Figure 7). Hence, P offers a platform onto which both proteins, M2-1 and PP1, can bind in close spatial proximity for M2-1 to be dephosphorylated by PP1. The PP1- and M2-1-binding motifs are nearly contiguous in the sequence of P and on the same domain, P_NT_. However, due to steric hindrance, binding of the two proteins probably occurs on two different protomers. This again underlines the essential role of the quaternary organization of the P protein that offers a structurally more complex environment than afforded by a single domain. Moreover, the proximity of the stably folded oligomerization may counterbalance the effect of the intrinsic disorder, which may be unfavorable to the formation of a ternary complex, by restricting the space that can be explored by the two SLiMs once they are bound to their respective bulky partner proteins.

## 7. Conclusions and Perspectives

*Pneumoviridae* P proteins are examples of atypical multidomain proteins. Due to disorder, structure determination of *Pneumoviridae* P has proven a complex endeavor. However, by adding pieces together, from isolated protein and from two-by-two protein complexes, a picture of the structure–function relationship in these proteins started to emerge, providing a structural basis to unravel the molecular mechanisms underlying the replication of pneumoviruses. *Pneumoviridae* P proteins act as the central architectural piece of the viral *holo* polymerase complex, where the polymerase is associated to its ribonucleoprotein template. They perform a structural role by providing a scaffold for several multi-protein complexes, reviewed herein. Their role as docking platforms arises from their organization into three domains, with a central folded oligomerization domain and two flanking regions. These are neither compact nor rigid, but still form distinct intrinsically disordered domains, P_NT_ and P_CT_.

It is remarkable that the only domain with a stable structure, P_OD_, arises from quaternary organization, i.e., from the tetramerization of a small region. On the other hand, disorder of P_NT_ and P_CT_ provides the necessary structural adaptability and several binding sites of different sizes. Notably, the two flanking domains, P_NT_ and P_CT_, have their own specificities. Whereas P_NT_ displays a large degree of disorder, P_CT_ displays higher order, even in the absence of binding partners. This suggests that these two domains are not equivalent in terms of the binding potential. P_NT_ contains several short linear binding sites. In the cases of M2-1 and RNA-N, the small size of the binding sites correlates with the weak affinity of the complexes formed with P, as each protein needs to dissociate from P to form new interactions, in a cyclic manner. In contrast, P_CT_ is nearly completely engaged in the L–P complex, which needs to be stable for a functional polymerase.

Interactome analysis has revealed that RSV P also recruits cellular factors to the polymerase complex, such as the Hsp70 chaperone, or associates with tropomyosin [94,95]. Binding regions for these proteins have not been specified yet, but could coincide with the previously identified potential binding regions of P. Other interaction sites might be detected by NMR interaction experiments that reveal changes in the dynamics of the disordered regions induced by binding. This has been done before for RSV P protein [68], but also for measles virus P protein, where ultraweak interactions, yet essential for viral replication, were characterized [96].

In conclusion, *Pneumoviridae* phosphoproteins illustrate the structural and dynamic versatility of multidomain proteins with large intrinsically disordered domains. These can be grasped by a multi-scale strategy, ranging from the molecular to atomic scale, which may be transposed to other proteins with similar structural organization and features. Regarding spectroscopic techniques, NMR and SAXS proved to be precious tools to grasp the dynamic properties of isolated protein. X-ray crystallography and cryogenic electron microscopy confirmed the stabilization of the transient structures by complex formation with protein partners. In the case of *Pneumoviridae* P proteins, it was essential to be able to work with protein fragments, which was facilitated by the disordered nature of the P_NT_ and P_CT_ domains.

## Figures and Tables

**Figure 1 ijms-22-01537-f001:**
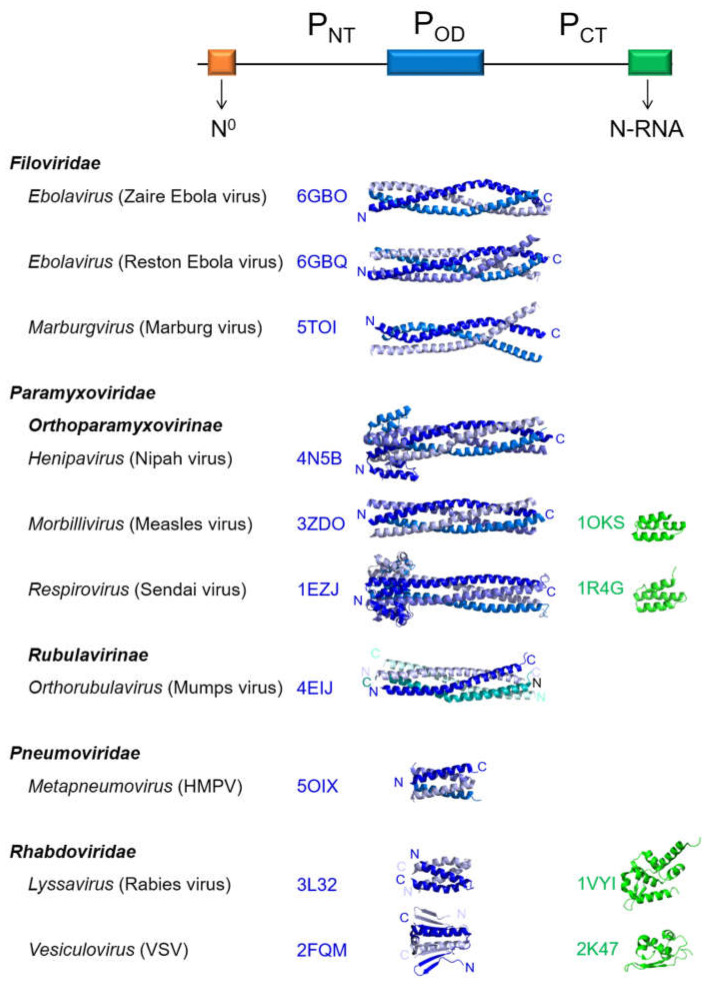
Overall architecture of the *Mononegavirales* phosphoproteins (P) and the structural diversity of the individual domains. On top is shown a scheme of the P protein, with the 3-fold repartition between the *N*- and *C*-terminal regions (P_NT_ and P_CT_), comprising distinct binding sites for the nucleoprotein (N), either as monomeric N^0^ or as an N-RNA complex, separated by an oligomerization domain (P_OD_). Representative structures are shown below, in cartoon, for P_OD_ (protomers are in different shades of blue) and the *C*-terminal N-RNA-binding domains (green), for viruses belonging to the *Filoviridae*, *Paramyxoviridae*, *Pneumoviridae* and *Rhabdoviridae* families. PDB codes are indicated next to the structure. Sub-families and genera are indicated in bold and regular font, respectively. Virus names are given in parentheses. *N*- and *C*-termini are indicated by N and C letters. For parallel oligomers, this indication is given only for the first protomer.

**Figure 2 ijms-22-01537-f002:**
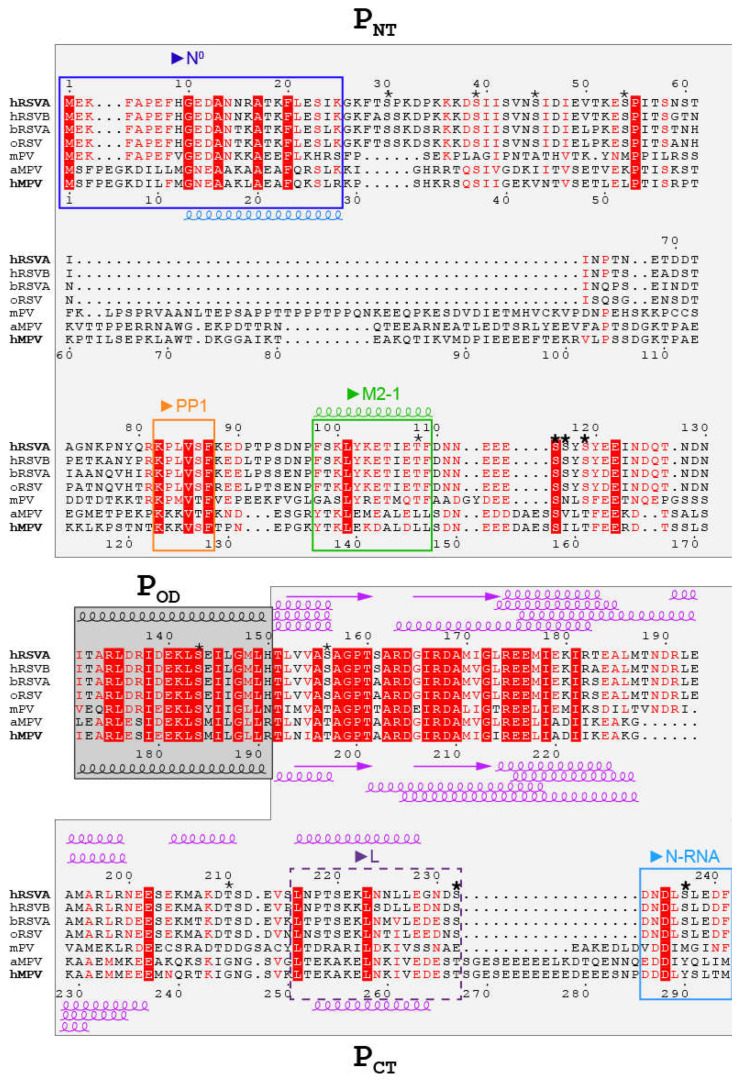
Sequence alignment of the *Pneumoviridae* phosphoproteins for the two genera *Orthopneumovirus*: human RSV strains A and B (hRSVA/hRSVB, Uniprot record numbers P03421/O42062), bovine RSV (bRSV, P33454), ovine RSV (oRSV, Q83956) and murine pneumonia virus (mPV, Q5MKM7); and *Metapneumovirus*: avian MPV (aMPV, Q2Y2M5) and human MPV (hMPV, Q8B9Q8). Sequence numbers are indicated for hRSVA on top and for hMPV at the bottom. Coloring was made according to sequence conservation with ESPript 3 [35]. Constitutively phosphorylated hRSV P serines are marked with bold stars. Other phosphorylatable Ser/Thr sites are marked with regular stars. Secondary structures, drawn for hRSVA and hMPV P, were taken from the X-ray crystal structures of hMPV P_OD_ (PDB 5oix), hMPV N^0^–P (PDB 5fvd) and hRSV M2-1–P (PDB 6g0y) and cryo-EM structures of hRSV L–P (PDB 6pzk) and hMPV L–P (PDB 6u5o) complexes. P_CT_ protomers adopt different conformations in the L–P complex. The interaction regions are boxed, and the protein partners indicated on top. For L, only the partial binding region reported in [36] is shown.

**Figure 3 ijms-22-01537-f003:**
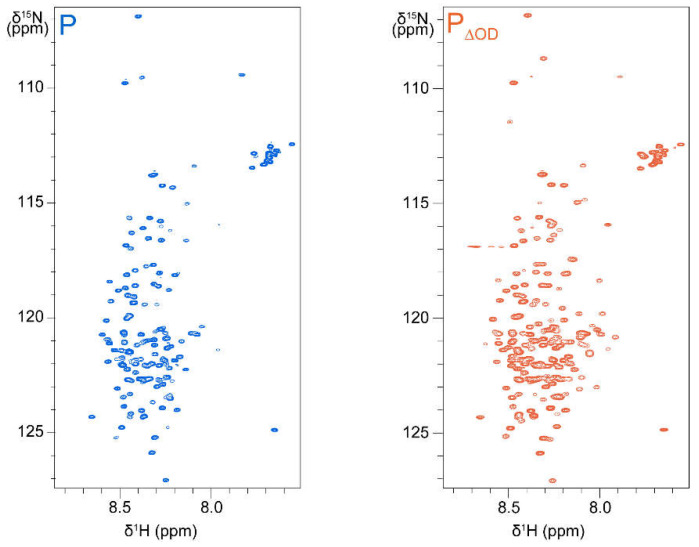
2D ^1^H–^15^N HSQC spectra of full-length RSV P and P_ΔOD_, a deletion mutant of P devoid of the oligomerization domain. Acquisition was done at a temperature of 288 K and at a magnetic field of 14.T (600 MHz ^1^H resonance frequency), using a triple resonance cryoprobe equipped with Z-axis gradients. Samples contained 175 µL of protein at concentrations of 250 µM for P and 80 µM for P_ΔOD_, in a 20 mM Na phosphate 100 mM NaCl buffer at pH 6.5, and 15 µL of D_2_O to lock the NMR spectrometer frequency. Assignments are not indicated, but can be retrieved from [68].

**Figure 4 ijms-22-01537-f004:**
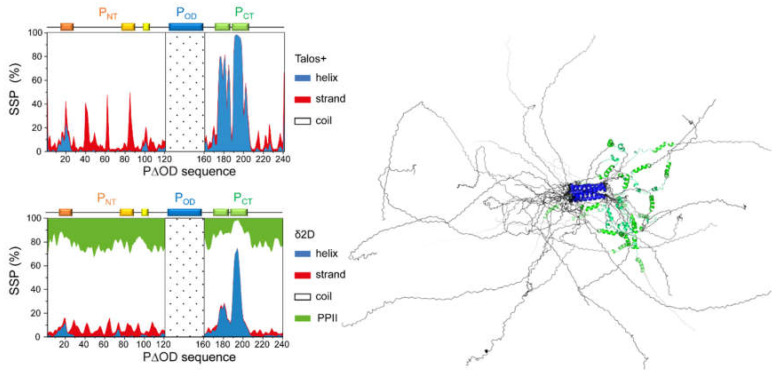
Secondary structure propensities for an RSV P construct deleted of the oligomerization domain (RSV P_ΔOD_), predicted from backbone chemical shifts, using two different algorithms, Talos+ [69] and δ2D [70]. The dotted area on the diagrams marks the deleted region in P_ΔOD_. On top of the diagrams are indicated the locations in the sequence of RSV P of P_OD_ and of regions with α-helical propensity or increased ^15^N relaxation (see Figure 5). On the right are shown structural models (5 tetramers) of RSV P, calculated with CYANA software [71] using only the torsion angle constraints derived from the backbone chemical shifts of P_ΔOD_ and P_OD_.

**Figure 5 ijms-22-01537-f005:**
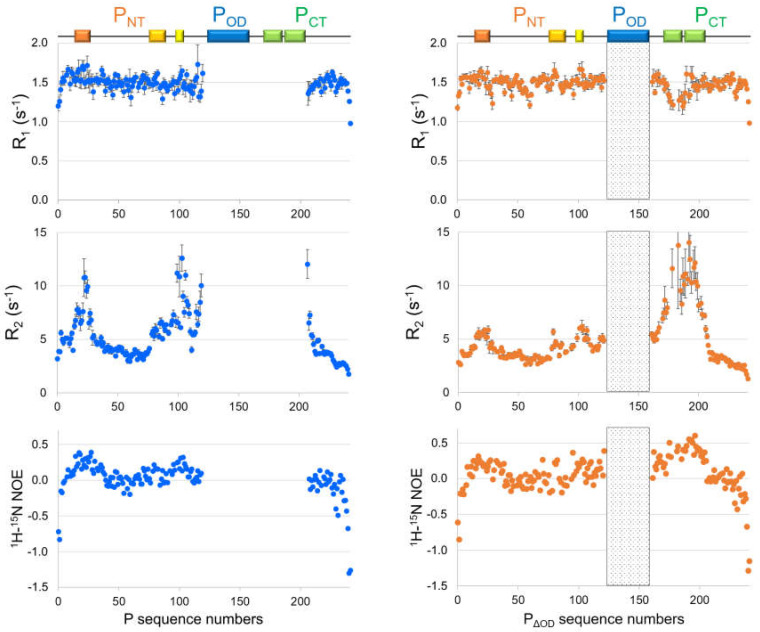
Comparison of the ^15^N spin relaxation properties between full-length RSV P and P_ΔOD_, a P fragment deleted of the oligomerization domain (indicated with dotted boxes on the diagrams for P_ΔOD_ data). The ^15^N nuclear longitudinal and transverse relaxation rates, R_1_ and R_2_, were plotted as well as the heteronuclear Overhauser effect, determined for each backbone amide signal along the protein sequence. Measurements were done at a temperature of 288 K, at a magnetic field of 14.T, using a triple resonance cryoprobe equipped with Z-axis gradients. On top are highlighted the regions in the sequence of RSV P with increased R_2_ values, indicative of either structured regions or regions in conformational exchange between disordered and folded forms.

**Figure 6 ijms-22-01537-f006:**
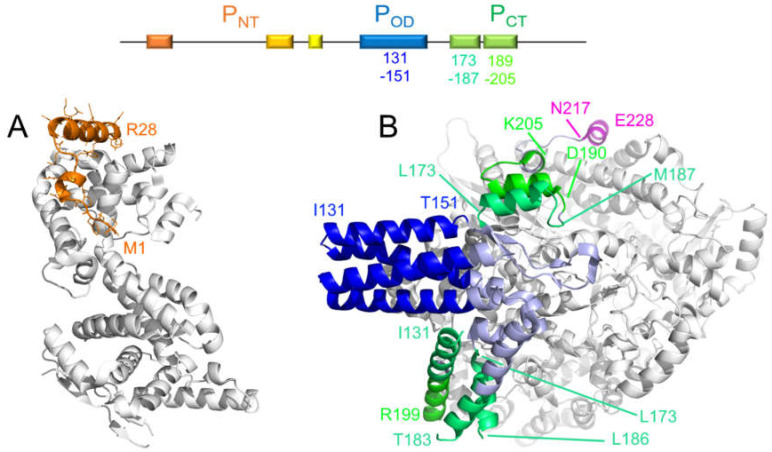
Folding of *Pneumoviridae* P protein upon binding to N^0^ and L proteins. (**A**) X-ray crystal structure of the hMPV N^0^–P complex, with stabilization of a transient *N*-terminal helix of P (PDB code 5FVD) [60]. (**B**) Structural model of the RSV L–P complex determined by cryogenic electron microscopy at a 3.2 Å resolution (PDB 6PZK) [32]. A similar structural organization was observed in a second cryo-EM structure of the RSV L–P complex at a 3.67 Å resolution (PDB 6UEN) [33] as well as in an hMPV L–P complex at a 3.8 Å resolution (PDB 6U5O) [34]. The partner proteins, N^0^ and L, are represented in white cartoon. The P fragments are shown in cartoon and in color. The color code for the complexed P helices is indicated in the scheme on top of the figure. In the L–P complex, the regions located between the tetramerization domain and the *C*-terminal transient helices predicted by NMR are shown in pale blue. The *C*-terminal helix N217-E228, which has no α-helical propensity in free P, is shown in magenta. The precise topology of each P protomer is indicated in Figure 2.

**Figure 7 ijms-22-01537-f007:**
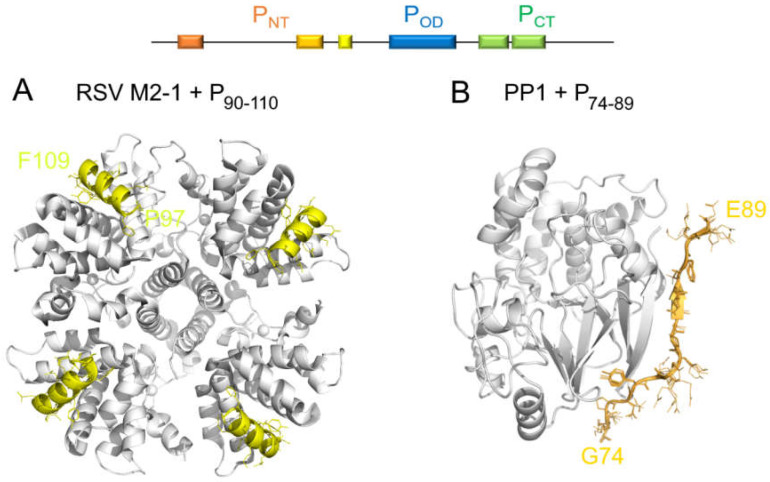
Stabilization of the structure of two short linear motifs (SLiMs) in RSV P protein, located in the *N*-terminal domain, upon complex formation. (**A**) X-ray structure of the hRSV M2-1–P complex, where a short transient helix, P residues P97–F109, is stabilized (PDB 6G0Y) [90]. (**B**) Homology model of the complex between the RVxF-like motif of P and cellular protein phosphatase 1, built with Modeller software [91]. The model was obtained by using the X-ray structure of the complex between the catalytic unit of PP1 and another PP1-binding protein, Gadd34 (PDB 4XPN), and by aligning the RVxF-like motifs of P and Gadd34. P fragments are shown in cartoon and lines. The color code is given in the scheme on top of the figure. The partner proteins are represented in the white cartoon.

## Data Availability

Data is contained within the article.

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
