# Peer review of "Pneumoviral Phosphoprotein, a Multidomain Adaptor-Like Protein of Apparent Low Structural Complexity and High Conformational Versatility"

_ijms, 2021, doi:10.3390/ijms22041537_

Round 1

Reviewer 1 Report

This review describes structural properties of unique class of proteins that are characterized by both disordered sections and conformational versatility making it really difficult to study. The review is well written, providing very good overview on the proteins, their function and the structure, and the methods used to study it. I am positive it will serve the community

Author Response

We thank the reviewer for her/his very positive report.

Reviewer 2 Report

The manuscript by Cardone et al. provides a thorough, in-depth, comprehensive review of pneumoviral phosphoproteins (P), which are multidomain cofactors of the viral polymerase, with very interesting and underexplored to date structural and dynamic features. The review focuses on the intrinsic disorder within these non-globular proteins, and shows compelling evidence that this feature is functional, i.e. that the combination of intrinsic disorder and tetrameric organisation of these proteins enable them to adapt to and interact with different partners, and to exert their function as adaptor-like platform. 

The manuscript is very well written (I believe I received the version that has been already editing, considering the highlighted regions and commented sections), with good quality figures and diagrams. It reads very well and I am happy to recommend the manuscript to be published as is, with only a very minor revision.  I would suggest to edit/rewrite the following sentences in the abstract:

All other parts are either flexible or form transient secondary structure elements that transiently associate with the rest of the protein.

The “transient” is redundant, perhaps replacing the first occurrence by “short-lived”, “temporary” or alike would work better. 

Author Response

We thank the reviewer for her/his very positive report. According to her/his remark, we changed "transient" into "short-lived" in the abstract (Page 1 line 18) to avoid a two-fold occurrence of "transient" in a same sentence.

This manuscript is a resubmission of an earlier submission. The following is a list of the peer review reports and author responses from that submission.

Round 1

Reviewer 1 Report

This review describes the viral phosphoprotein and its interaction with its partner proteins, and provides detailed structural information on this complex and function. 

The review is very comprehensive spanning all literature in this field. However, the methodologies are described in too detailed and I would recommend to shorten the entire review by removing/shortening these parts. Then for me the conclusions at the end of the paragraphs were missing. What do we gain from these insights? Why is it important? What are the next steps? It would be worth for the readers to provide also new directions in this field and not only a summary of what was achieved till now.

Author Response

We would like to thank the reviewer to have read through the manuscript and provided guidelines to improve it.

Following the reviewer's advice, we have shortened the methodology part (paragraph 3.3 in particular) by suppressing a number of technical details to go straight to the structural interpretation.

We added a short introduction, rewrote the conclusions, and made modifications in the abstract, to situate this review in the broader context of multidomain proteins, stressing the particular feature of intrinsically disordered domains. We also indicated, which questions should be and can be addressed next, building on the knowledge acquired on the family of proteins presented in this review.

All modifications were made using track changes in the text.

Reviewer 2 Report

.The manuscript is complex and difficult to follow for someone outside these proteins. It is also full of abbreviations that make reading even thornier. I am not able to guess what the real purpose of this review is, apart from collecting a multitude of bibliographic details related to structural and functional studies. I miss a line of explanation that is much more didactic and with a more defined objective.

Author Response

We would like to thank the reviewer to have read through the manuscript and provided guidelines to improve it.

First, we removed unnecessary abbreviations, hoping that the text is now more readable. However, we kept some abbreviations that were used more frequently, either in a same paragraph or throughout the text.

Following the reviewer's advice, we added a short introduction, rewrote the conclusion, and made modifications in the abstract, to situate this review in the broader context of multidomain proteins, stressing on the particular feature of intrinsically disordered domains, as compared to globular domains. We hope that it is clearer now that our aim was to illustrate this topic through the structural and functional implications of this kind of domains for a specific family of proteins, presented in this review.

Since this review focuses on a particular family of proteins, we also added perspectives to the conclusion, indicating which questions should be addressed next, building on the previously acquired knowledge.

All modifications were made using track changes in the text.

Round 2

Reviewer 2 Report

I find this manuscript very descriptive at an experimental level but difficult to follow by, for instance, a structural biologist who wants to learn about these proteins. Thus, and especially in the first two sections, during the speech the structural details are mixed with many other functional and experimental ones that dilute and distract from the relevant information. In addition, numerous partners accumulate, with their names, functions and structural characteristics, which are not more relevant later, but they blur the ability to extract clear information for someone outside these systems. More concretely, at section 2, the authors give first a few structural details, which I think is fine, and then a descriptive of a lot of possible partners of the P protein, but nothing structural, which is uninformative and away from intention.